# Reverse Genetics of Murine Rotavirus: A Comparative Analysis of the Wild-Type and Cell-Culture-Adapted Murine Rotavirus VP4 in Replication and Virulence in Neonatal Mice

**DOI:** 10.3390/v16050767

**Published:** 2024-05-12

**Authors:** Takahiro Kawagishi, Liliana Sánchez-Tacuba, Ningguo Feng, Harry B. Greenberg, Siyuan Ding

**Affiliations:** 1Department of Molecular Microbiology, Washington University School of Medicine, St. Louis, MO 63110, USA; 2Department of Medicine, Division of Gastroenterology and Hepatology, Stanford University School of Medicine, Stanford, CA 94305, USA; 3Department of Microbiology and Immunology, Stanford University School of Medicine, Stanford, CA 94305, USA; 4VA Palo Alto Health Care System, Department of Veterans Affairs, Palo Alto, CA 94304, USA

**Keywords:** rotavirus, reverse genetics, small-animal model

## Abstract

Small-animal models and reverse genetics systems are powerful tools for investigating the molecular mechanisms underlying viral replication, virulence, and interaction with the host immune response in vivo. Rotavirus (RV) causes acute gastroenteritis in many young animals and infants worldwide. Murine RV replicates efficiently in the intestines of inoculated suckling pups, causing diarrhea, and spreads efficiently to uninoculated littermates. Because RVs derived from human and other non-mouse animal species do not replicate efficiently in mice, murine RVs are uniquely useful in probing the viral and host determinants of efficient replication and pathogenesis in a species-matched mouse model. Previously, we established an optimized reverse genetics protocol for RV and successfully generated a murine-like RV rD6/2-2g strain that replicates well in both cultured cell lines and in the intestines of inoculated pups. However, rD6/2-2g possesses three out of eleven gene segments derived from simian RV strains, and these three heterologous segments may attenuate viral pathogenicity in vivo. Here, we rescued the first recombinant RV with all 11 gene segments of murine RV origin. Using this virus as a genetic background, we generated a panel of recombinant murine RVs with either N-terminal VP8* or C-terminal VP5* regions chimerized between a cell-culture-adapted murine ETD strain and a non-tissue-culture-adapted murine EW strain and compared the diarrhea rate and fecal RV shedding in pups. The recombinant viruses with VP5* domains derived from the murine EW strain showed slightly more fecal shedding than those with VP5* domains from the ETD strain. The newly characterized full-genome murine RV will be a useful tool for dissecting virus–host interactions and for studying the mechanism of pathogenesis in neonatal mice.

## 1. Introduction

Rotavirus (RV) is the most common causative agent of severe acute diarrhea in infants and small animals worldwide [1]. While RV has been isolated from many mammalian and avian species, RV infection relatively infrequently demonstrates cross-species transmission and persistence in a heterologous host species, a phenomenon known as host-range restriction (HRR) [2,3,4,5,6,7,8,9,10]. For instance, the genome sequences of RV strains isolated from humans generally belong to groups of human RV strains isolated before. RVs from other animal species are occasionally isolated from humans but rarely spread or persist in the human population [11]. HRR has been exploited to generate two live-attenuated RV vaccines currently used worldwide (i.e., RotaTeq (Merck) and RotaSiil (Serum Institute of India)), using bovine RV strains as a genetic backbone [12].

Since HRR contributes to natural attenuation, the molecular basis of RV HRR has been studied in animal infection models, especially in mice [2,3,6,10,13,14]. Homologous murine RV strains replicate and spread efficiently in the mouse model; however, heterologous RV strains (e.g., simian, bovine, porcine, and human RVs) do not. The RV genome consists of 11 segmented double-stranded RNAs that encode six structural proteins (VP1 to VP4, VP6, and VP7) and six nonstructural proteins (NSP1 to NSP6). Previous studies, including research by our group and others, have demonstrated that gene segments encoding VP3, VP4, VP7, NSP1, NSP2, NSP3, and NSP4 proteins can all be associated with RV HRR [5,7,9,10,13,14].

A natural mono-reassortant RV D6/2 strain was isolated by plaque assays from an intestinal homogenate of a suckling mouse co-infected with wild-type murine EDIM-EW and the tissue-culture-adapted simian RRV strain [5]. Unlike wild-type murine RV, D6/2 replicates in cultured cell lines while still efficiently causing diarrhea in inoculated pups [5,10]. Of note, 10 of the 11 gene segments of D6/2 are derived from the EDIM-EW strain; for the exception is gene segment 4, which encodes the cell attachment protein VP4 (Table 1). It has been shown that cell-culture-adapted murine RV strains do not cause diarrhea as efficiently as the wild-type murine RV [6,15,16,17]. Therefore, D6/2 provides a unique opportunity to further interrogate the viral determinants of HRR.

Since the first plasmid-based reverse genetics was developed for simian SA11 strain [18], reverse genetics has been established for several other animal RV strains [19,20,21,22,23,24,25,26]. We previously used D6/2 as a genetic backbone and rescued a recombinant murine-like RV by introducing two more gene segments from the simian SA11 strain: gene segments 1 and 10, which encode viral polymerase VP1 and viral enterotoxin NSP4, respectively (Table 1) [21]. The new recombinant virus D6/2 with the two additional genes derived from simian RV (so-called rD6/2-2g) replicated in the intestine of inoculated pups and transmitted to their uninoculated littermates. Using rD6/2-2g as the backbone, we have since demonstrated that murine RV NSP1, an interferon antagonist, plays a critical role in viral replication in vivo [27]. In addition, by comparing VP4s from heterologous RV strains in an isogenic rD6/2-2g background, we demonstrated that VP4s from heterologous RV strains contribute to HRR to varying degrees [14]. Furthermore, we rescued an rD6/2-2g RV expressing a bioluminescent reporter, Nano-Luciferase, to characterize systemic dissemination of RV in vivo in a non-invasive manner [28].

Despite the utility of rD6/2-2g, this recombinant murine-like RV is a compromise between rescue efficiency and in vivo virulence. It still harbors three gene segments (gene segments 1, 4, and 10) from heterologous simian RV strains (Table 1). Considering that these three gene segments may be potentially associated with HRR in mice, it is desirable to create an RV that grows well in culture but has all 11 gene segments derived from a murine strain to be used in a mouse model. Toward that objective, we attempted to replace the three heterologous gene segments in rD6/2-2g with those from a murine-origin RV strain.

The obvious primary challenge for this objective is the RV structural protein VP4. It is the spike protein present on the surface of RV virions and is required for cell attachment and entry in host cells. VP4 is cleaved by trypsin into two distinct domains (Figure 1A). The N-terminal VP8* domain forms the head structure of the virion spike and engages in attachment to the target cell surface. The β-barrel domain in VP5* forms the body of the spike, and the C-terminal region of the VP5* domain functions as the foot of the spike by interacting with VP7 and VP6 proteins in the virion [29,30,31]. Here, we used reverse genetics and improved the virulence of rD6/2-2g by rescuing a series of recombinant RVs with all 11 gene segments from a murine RV strain. We also generated murine RV VP4 chimeric viruses between the cell-culture-adapted ETD_822 strain and the wild-type murine EW strain and compared diarrheal diseases in the suckling mouse model. The data suggest that murine RV reverse genetics offers a new tool to study molecular mechanisms of RV replication, virulence, and spread in the homologous murine model.

## 2. Materials and Methods

### 2.1. Cells and Viruses

Monkey kidney MA104 cells (ATCC CRL-2378.1) were grown in Medium 199 (Gibco) supplemented with 10% fetal bovine serum (FBS), 2 mM L-glutamine, 100 I.U./mL penicillin, and 100 μg/mL streptomycin. Baby hamster kidney cells constitutively expressing T7 RNA polymerase (BHK-T7 cells) were kindly gifted by Dr. Buchholz at the NIH [32] and cultured in Dulbecco-modified essential medium (DMEM) (CORNING) supplemented with 10% FBS, L-glutamine (2 mM), penicillin (100 I.U./mL), and streptomycin (100 μg/mL). The cells were cultured in the presence of Geneticin (1mg/mL) every other passage to maintain the clone expressing T7 RNA polymerase. The natural reassortant D6/2 was generated previously [5] and was propagated in MA104 cells. Briefly, the virus was activated by 5 μg/mL of trypsin (Sigma-Aldrich, St. Louis, MO, USA) at 37 °C for 15 min and propagated in MA104 cells in serum-free Medium 199 (SFM) with 0.5 μg/mL of trypsin.

### 2.2. Plasmid Construction

To construct rescue plasmids for D6/2 gene segments 1 (VP1) and 10 (NSP4), viral dsRNAs were extracted from freeze-thawed stock of D6/2 with Trizol Reagent (Thermo Fisher Scientific, Waltham, MA, USA). Viral gene segments were determined as previously described [33]. Briefly, a self-anchoring primer was ligated to the 3′ termini of viral dsRNAs with T4 RNA Ligase (New England Biolabs, Ipswich, MA, USA), and then viral cDNAs were synthesized with High-Capacity cDNA Reverse Transcription Kit (Applied Biosystems, Waltham, MA, USA). Gene segments 1 and 10 were amplified by PrimerSTAR HS DNA Polymerase (Takara Bio, San Jose, CA, USA) and used to replace the cDNA of SA11 NSP4 in the pT7-SA11-NSP4 plasmid by the NEBuilder HiFi DNA Assembly Master Mix (New England Biolabs, Ipswich, MA, USA). Nine rescue plasmids for D6/2 (pT7-D6/2-VP2, -VP3, -VP4, -VP6, -VP7, -NSP1, -NSP2, -NSP3, and -NSP5) and the rescue plasmid encoding ETD_822-VP4 (pT7-ETD_822-VP4) were previously constructed [14,21]. To generate the rescue plasmid for EW VP4 (pT7-EW-VP4), the cDNA (GenBank accession number: U08429) was synthesized and cloned between the T7 promoter and HDV ribozyme sequences in the pT7-SA11-NSP4 plasmid by replacing the cDNA of SA11-NSP4 with that of EW-VP4. To generate VP4 chimeric plasmids between the ETD_822 and EW strains (pT7-ETD-VP4-EW-VP8*, pT7-ETD-VP4-EW-VP5*-body, and pT7-ETD-VP4-EW-VP5*-foot), sequences for nucleotides 1 to 1000, 1001 to 1500, and 1500 to 2331 in pT7-ETD_822-VP4 were replaced with those for EW VP4.

### 2.3. Reverse Genetics

Recombinant viruses were generated using an optimized reverse genetics protocol, as reported previously [21]. Briefly, we mixed 11 rescue plasmids and one helper plasmid (0.4 μg for nine of the rescue plasmids (excluding for NSP2 and NSP5), 1.2 μg of the two rescue plasmids for NSP2 and NSP5, and 0.8 μg of C3P3-G1) in OPTI-MEM I Reduced-Serum Medium (Thermo Fisher Scientific, Waltham, MA, USA). The mixture of the plasmids was transfected into BHK-T7 cells using TransIT-LT1 (Mirus, Madison, WI, USA). The next day, the medium was replaced with serum-free DMEM and cultured overnight. Then, MA104 cells were added to the BHK-T7 cells and cultured in the presence of 0.5 μg/mL of trypsin. To generate the VP4 chimeric viruses, we replaced the rescue plasmid for VP4 with the appropriate plasmids. Rescued viruses were amplified in MA104 cells, and the VP4 sequence of the viruses was confirmed by DNA sequencing before use.

### 2.4. Focus-Forming Unit Assays

MA104 cells were seeded on 96-well plates and cultured for 2 to 3 days. Virus samples were activated with 5 μg of trypsin, serially diluted with SFM, and inoculated into MA104 cells. The cells were fixed with 10% formalin (Fisherbrand Waltham, MA, USA) 14 h after inoculation, permeabilized with PBS with 0.05% Triton X-100, and stained with rabbit anti-RV DLP and HRP-conjugated anti-rabbit IgG polyclonal antibody (Sigma-Aldrich, St. Louis, MO, USA). RV antigen was visualized with the AEC Substrate Kit and peroxidase (Vector Laboratories, Newark, CA, USA). The number of foci was counted under a microscope, and the virus titer was expressed as FFU/mL.

### 2.5. Mouse Infection

129sv mice were purchased from Taconic Biosciences Inc. and maintained at the animal facility in the Veterinary Medical Unit of the Palo Alto VA Health Care System. Five-day-old 129sv pups were orally inoculated by gastric lavage with 1 × 10^3^ FFU of recombinant viruses or 1 × 10^3^ DD_50_ of the wild-type EW strain. Mice were monitored to collect stool samples by gentle abdominal pressure for 12 days. Stool samples were collected in 40 μL of PBS (+) (CORNING) and stored at −80 °C until use. The animal experiment protocol was approved by the Stanford Institutional Animal Care Committee.

### 2.6. ELISA

The relative quantity of RV fecal shedding was assessed by sandwich ELISA, as previously described, using guinea pig anti-RV TLP antiserum and rabbit anti-RV DLP antiserum generated in the Greenberg lab [34]. Briefly, ELISA plates (E&K Scientific Products, Swedesboro, NJ, USA, cat. #EK-25061) were coated with guinea pig anti-RV TLP antiserum and blocked with PBS supplemented with 2% BSA. After washing the plate with PBS containing 0.05% Tween 20, 70 μL of PBS containing 2% BSA and 2 μL of the fecal samples were added to the plate and incubated at 4 °C overnight. The RV antigen in the stool samples was detected by rabbit anti-RV DLP antiserum, HRP conjugated anti-rabbit IgG (Sigma-Aldrich, St. Louis, MO, USA, cat. #A0545), and peroxidase substrate (SeraCare, Milford, MA, USA). The signal intensity at 450nm was measured with the ELx800 microplate reader (BIO-TEK, Shoreline, WA, USA).

### 2.7. Statistical Analysis

Fecal shedding curves by RVs were analyzed by two-way ANOVAs with the Tukey multiple comparison test using GraphPad Prism 8.

## 3. Results

### 3.1. Generation of Recombinant Murine RVs

In a previous study, we synthesized all 11 rescue plasmids from the D6/2 strain. However, we were unable to rescue a completely recombinant D6/2 strain after multiple trials [21]. Therefore, we used recombinant D6/2 with gene segments 1 and 10, which encode VP1 and NSP4, from the simian SA11 strain as an alternative approach [21]. The amino acid sequence identity of VP1 and NSP4 between the murine EW and the simian SA11 strains showed 86.2% and 62.3% homology, respectively. To generate a recombinant virus with a gene constellation closer to a fully murine RV, we reconstructed rescue plasmids for gene segments 1 and 10 from the D6/2 strain. To our surprise, we obtained recombinant D6/2 (rD6/2) with the new rescue plasmids, despite there being no difference in the cDNA sequences of gene segments 1 and 10 compared with those in the previous failed rescue plasmids.

We next attempted to further optimize gene segment 4 in rD6/2, which is derived from the simian RRV strain. Wild-type murine RV strains (including the EW strain) propagated in mouse intestines do not efficiently infect immortalized cell lines. Previous studies that compared the nucleotide sequences of murine RV before and after adaptation to cultured cells reported that gene segment 4 is one of the determinants for effective viral replication in cultured cell lines [17]. It suggests that murine RV from mouse intestines poorly replicates in the cell line possibly due to a partial restriction at the attachment and entry process. Therefore, we used the rescue plasmid for gene segment 4 of the cell-culture-adapted murine ETD_822 strain, and we generated a recombinant virus with 10 gene segments from murine EW and gene segment 4 from the ETD_822 strain (rEW/ETD-VP4) (Table 1).

We compared the nucleotide sequence of VP4 between the EW and ETD_822 strains to better understand the difference in VP4 in the murine RV strain used in this study. Sequence alignment shows that compared with the wild-type EW strain, ETD_822 has only five non-synonymous amino acid substitutions (Y80H, D452N, S470L, T612A, and A711T) in VP4. Y80H is the only amino acid difference found in the VP8* domain, and the VP5* domain has two amino acid differences in either the body (D452N and S470L) or the C-terminal foot (T612A and A711T) regions (Figure 1B). It is known that the cell-culture-adapted EDIM murine RV strains are attenuated in suckling mice in terms of diarrheal dose and duration of shedding while having acquired the ability to replicate in cultured cell lines [6,15,16,17]. We speculated that some amino acids are strongly associated with the adaptation to cultured cell lines, but not all amino acids are necessary for efficient replication in cell lines. To test whether we could rescue a recombinant RV with a VP4 protein closer to the more virulent, non-cell-culture-adapted progenitor EW strain, we constructed three VP4 chimeric plasmids between the ETD_822 and EW strains. These plasmids harbor nucleotide sequences of the VP8*, VP5*-body, or VP5*-foot domains from the EW strain in ETD_822 VP4 (Figure 1B). Of note, we successfully rescued all three chimeric viruses, namely rEW/ETD-VP4-EW-VP8*, rEW/ETD-VP4-EW-VP5*-body, and rEW/ETD-VP4-EW-VP5*-foot. These data suggest that amino acid differences in these three regions in ETD_822 are not individually involved in the adaptation to cultured cell lines.

### 3.2. Comparative Analysis of Diarrhea Rate by Recombinant Murine RVs in a Suckling Mouse Model

To assess the capacity of the rescued viruses to induce diarrhea, we inoculated 5-day-old 129sv pups with 1 × 10^3^ FFU of the recombinant murine RVs and 1 × 10^3^ DD_50_ of the highly virulent non-cell-culture-adapted murine RV EW strain as a control. We monitored the mice for 12 days to compare the percentage and duration of diarrhea occurrence. The wild-type, non-cell-culture-adapted murine EW strain caused 100% diarrhea in all inoculated pups from 2 to 9 days post-inoculation (Figure 2A). Compared with EW, rD6/2 was slightly attenuated and did not cause diarrhea in all the pups (Figure 2B), consistent with the previous literature [10]. The new rEW/ETD-VP4 virus had a similar disease phenotype in that it caused diarrhea but not in all pups over time (Figure 2C), suggesting that ETD VP4 is not more virulent than RRV VP4. The three VP4 chimeric viruses (rEW/ETD-VP4-EW-VP8*, rEW/ETD-VP4-EW-VP5*-body, and rEW/ETD-VP4-EW-VP5*-foot) caused diarrhea in inoculated pups but did not show a diarrhea phenotype as robust as that by murine EW (Figure 2D–F). The data suggest that VP8* or the body or foot regions of VP5* from the EW strain did not individually increase the diarrheal rates compared with the parental virus with ETD-VP4 (rEW/ETD-VP4) in the suckling mouse model.

### 3.3. Comparative Analysis of Fecal RV Shedding by Recombinant Murine RVs in a Suckling Mouse Model

Next, we compared the amount of fecal RV shedding among the various VP4 constructs. Wild-type murine RV caused a curve with a single peak of more than 2.0 at OD_450_ at 4 days post-inoculation, demonstrating robust replication in the mouse intestine (Figure 3A). In contrast, the fecal RV shedding curve from the rD6/2-inoculated pups showed two peaks on days 2 and 6 post-inoculation, and the OD values did not reach as high as those of the EW strain (Figure 3B). The other four viruses that had the murine RV VP4 gene demonstrated three peaks on day 2, from day 5 to day 7, and from day 10 to 11 days post-inoculation, and none of these viruses reached the high levels of fecal shedding as seen with the wild-type EW strain (Figure 3C–F). Statistical analysis of the fecal RV shedding between the recombinant viruses and the EW strain confirmed that none of the recombinant viruses were shed to the same level as that by the wild-type murine EW strain (Table 2). We also found that, compared with rEW/ETD-VP4, two of the three VP4 chimeras, i.e., rEW/ETD-VP4-EW-VP5*-body and rEW/ETD-VP4-EW-VP5*-foot, caused more fecal RV shedding, whereas rEW/ETD-VP4-EW-VP8* did not (Table 2). These results suggest that, among the five different amino acids in VP4 between the EW and ETD_822 strains, amino acids in the VP5* region are positively associated with efficient replication in the mouse intestine.

## 4. Discussion

In this study, we leveraged an optimized reverse genetics system to improve the virulence of the murine RV rD6/2-2g strain by exchanging the remaining three gene segments from heterologous simian SA11 or RRV strains with its homologous murine counterparts and rescued a recombinant RV with 11 gene segments all derived from a murine RV strain (rEW/ETD-VP4). We previously attempted to rescue rD6/2 with rescue plasmids of gene segments 1 and 10 constructed by DNA synthesis. After constructing the plasmids, we performed reverse genetics with different clones and repeated this multiple times; however, none of the rescue experiments were successful. In the current study, we constructed the plasmids again by cloning the gene segments from the original D6/2 stock. Of note, the new plasmid sequences of the T7 promoter, RV cDNA, hepatitis delta virus ribozyme, and T7 terminator, although identical to the original plasmids, led to the successful rescue of rD6/2. It suggests that clonal differences might affect the rescue efficiency in reverse genetics. It is uncertain whether there is a difference in some other parts of the plasmid, and, if that is the case, whether this affects the reverse genetics results. Whole-plasmid sequencing of the plasmids would be helpful to examine whether there is any difference between the clones. It would be important to test multiple clones prepared separately when some rescue plasmids do not work, even if the plasmid has the correct sequence.

We replaced three gene segments, which encode the RNA-dependent RNA polymerase VP1 (encoded by gene segment 1), the cell attachment protein VP4 (encoded by gene segment 4), and the viral enterotoxin NSP4 (encoded by gene segment 10). Among these gene segments, gene segment 4 has been implicated in RV HRR; however, the contribution of gene segments 1 and 10 to HRR is less clear. VP1 interaction with VP2 is critical for transcription and genome replication [35]. Group A RVs have 28 VP1 genotypes and 24 VP2 genotypes (Rotavirus Classification Working Group: RCWG updated on April 3rd 2023 (https://rega.kuleuven.be/cev/viralmetagenomics/virus-classification/rcwg)) [36,37], and it is reported that the combination of VP1 and VP2 genotypes changes the VP1 polymerase activity in some cases [38]. RV NSP4 is an enterotoxin that increases host calcium levels in the cytoplasm and activates calcium-ion-dependent chloride channels, and it is directly involved in causing diarrhea [39]. In light of the sequence differences between the EW and SA11 strains, we preferred using gene segments 1 and 10 originating from murine RV to specifically focus on studying viral replication, virulence, and spread of murine RV in a mouse model.

In our previous study, we compared the role of VP8* and VP5* from heterologous RV strains in virus replication and diarrhea in a suckling mouse model. We generated VP8* and VP5* chimeric viruses between homologous ETD and heterologous bovine UK strains on an rD6/2-2g background [14]. The results showed that, in the case of comparison between homologous and heterologous VP4s, both VP8* and VP5* from ETD contributed to increased diarrhea in the suckling mouse model [14]. In the present study, we evaluated the role of VP8* and VP5* from murine RV strains in a murine RV backbone. This is important because we are now testing VP4 in a genetic backbone identical to the homologous murine RV backbone, as opposed to the murine-like condition used in the previous study. Despite the different genetic background, we came to the same conclusion that ETD VP4 is not more virulent than RRV VP4, suggesting that when ETD VP4 is not available, RRV VP4 can serve as a robust surrogate for in vivo studies. To delineate the contributions of VP8* versus VP5*, we generated VP4 chimeric viruses between a non-tissue-culture-adapted EW strain and a tissue-culture-adapted ETD_822 strain and compared the role of VP8* and VP5* in a homologous murine RV strain. Of interest, VP4 chimeric viruses with VP5* body or foot regions, but not VP8*, slightly increased the amount of RV shedding in the feces compared with a control virus with ETD-VP4 (Figure 3C,E,F and Table 2). It is possible that the rEW/ETD-VP4-EW-VP5*-body and the rEW/ETD-VP4-EW-VP5*-foot replicate better than rEW/ETD-VP4 in MA104 cells. Previous studies on host factors involved with RV entry demonstrated that the VP8* domain of VP4 attaches to cell-surface glycans (e.g., sialic acid and histo-blood group antigens), while the VP5* domain interacts with other coreceptors (e.g., integrins and heat-shock cognate protein 70). Subsequently, VP5* likely plays a role in membrane penetration at a post-attachment step [31]. Our current results suggest that the difference in VP4 between non-tissue-culture-adapted EW and cell-culture-adapted ETD_822 occurs after the initial virion attachment step with cell-surface glycans. Of note, none of the recombinant viruses caused the same severe diarrheal diseases as EDIM-EW did (Figure 2). These data suggest that multiple mutations in VP4 or other viral proteins are required for robust replication in the mouse intestine.

One can imagine that there are multiple avenues available to leverage this powerful murine RV system to identify and study the molecular factors that modulate the severity of diarrhea and viral replication. For example, it would be interesting to further passage these recombinant viruses in mouse intestines, determine the nucleotide differences by next-generation sequencing, and introduce the mutations into the rescue plasmids to pinpoint the precise amino acids important for more robust replication in the mouse intestine without losing the ability of the virus to replicate in cultured cells. It would also be of interest to test these viruses in an adult mouse model to see if different results are obtained to those in the neonatal mouse system. Finally, although human enteroid cultures have proven a great tool for modeling primary human intestinal epithelial cells and for studying RV infection [40,41,42], such a system is lacking for the murine enteroids, which would be useful for teasing apart the stage of entry affected by VP8* and/or VP5* mutations. In conclusion, we have developed a reverse genetics for murine RV. This system will provide a useful tool for understanding the biology of RV in mouse models.

## Figures and Tables

**Figure 1 viruses-16-00767-f001:**
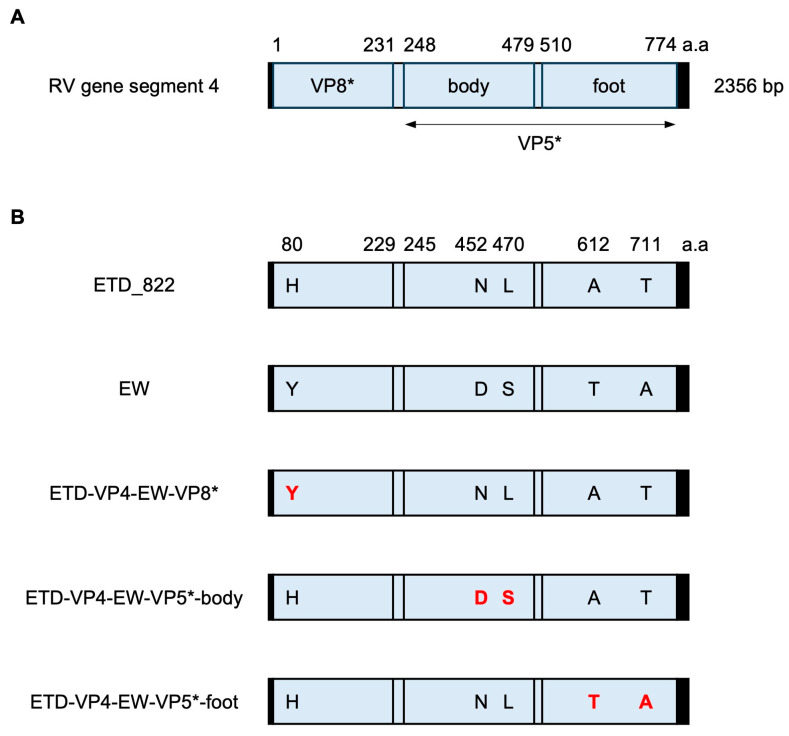
Schematic presentation of the murine RV VP4 gene. (**A**) Schematic presentation of RV gene segment 4. The 5′ and 3′ UTRs are shown as black boxes. VP8* and the body and foot regions of the VP5* domain in the VP4 gene are shown in light blue boxes. The numbers above the box indicate the amino acid positions. (**B**) Schematic presentation of the murine RV ETD_822 and EW strains and the VP4 chimeric viruses generated in this study. The five amino acids that differ between the ETD_822 and EW strains are highlighted in red inside the blue boxes. The number above the box indicates the amino acid positions.

**Figure 2 viruses-16-00767-f002:**
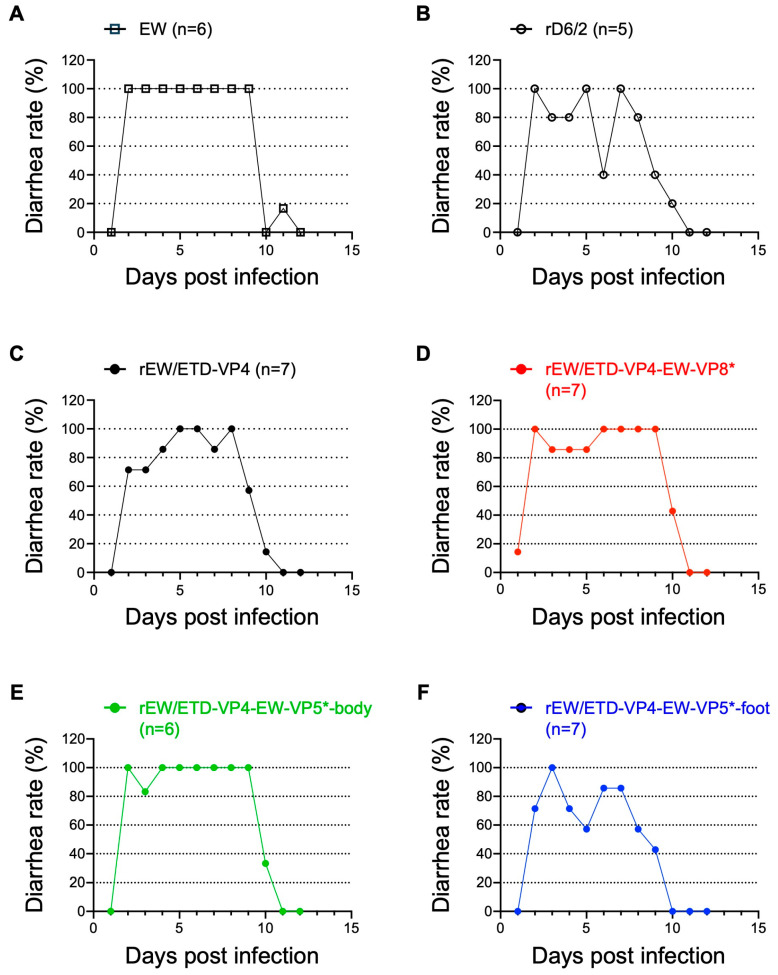
Percentage of diarrhea caused by the wild-type murine RV and recombinant murine RVs. Five-day-old 129sv pups were inoculated with (**A**) 1 × 10^3^ DD_50_ of EW, or 1 × 10^3^ FFU of (**B**) rD6/2, (**C**) rEW/ETD-VP4, (**D**) rEW/ETD-VP4-EW-VP8*, (**E**) rEW/ETD-VP4-ETD-VP5*-body, or (**F**) rEW/ETD-VP4-ETD-VP5*-foot. The infected mice were monitored for diarrheal stool for 12 days by gentle abdominal pressure.

**Figure 3 viruses-16-00767-f003:**
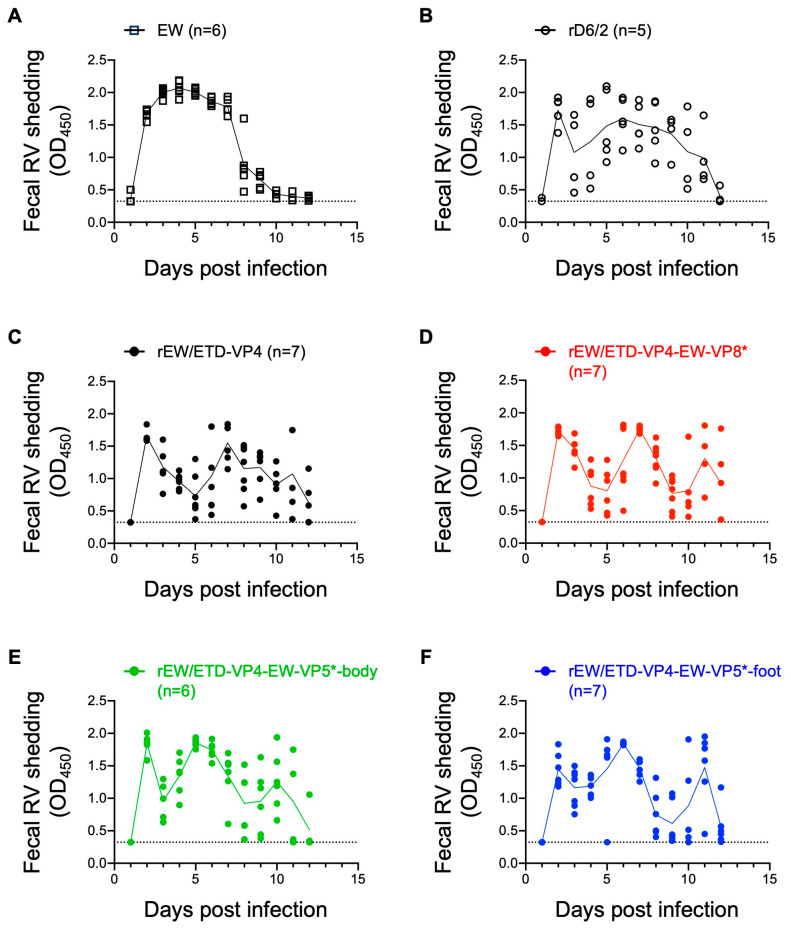
Fecal RV shedding by wild-type murine RV and recombinant murine RVs. Five-day-old 129sv pups were inoculated with the same doses and viruses as in Figure 2. (**A**) 1 × 10^3^ DD_50_ of EW, or 1 × 10^3^ FFU of (**B**) rD6/2, (**C**) rEW/ETD-VP4, (**D**) rEW/ETD-VP4-EW-VP8*, (**E**) rEW/ETD-VP4-ETD-VP5*-body, or (**F**) rEW/ETD-VP4-ETD-VP5*-foot. The amount of RV in the stool samples was determined by ELISA. Each dot shows data from one pup and the line shows the average score. The dotted lines indicate the score of the limit of detection determined from the stool of uninfected pups.

**Table 1 viruses-16-00767-t001:** Gene constellation of the D6/2 and recombinant viruses.

Gene Segment (Viral Protein)	D6/2	rD6/2-2g	rEW/ETD-VP4
Gene segment 1 (VP1)	EW	SA11	EW
Gene segment 2 (VP2)	EW	EW	EW
Gene segment 3 (VP3)	EW	EW	EW
Gene segment 4 (VP4)	RRV	RRV	ETD
Gene segment 5 (NSP1)	EW	EW	EW
Gene segment 6 (VP6)	EW	EW	EW
Gene segment 7 (NSP3)	EW	EW	EW
Gene segment 8 (NSP2)	EW	EW	EW
Gene segment 9 (VP7)	EW	EW	EW
Gene segment 10 (NSP4)	EW	SA11	EW
Gene segment 11 (NSP5/6)	EW	EW	EW

**Table 2 viruses-16-00767-t002:** Summary of the statistical analysis of fecal RV shedding ^1,2^.

Virus	Fecal RV Shedding (Compared with EW)	Fecal RV Shedding (Compared with rEW/ETD-VP4)
EW	n.a.	***
rEW/ETD-VP4	***	n.a.
rEW/ETD-VP4-EW-VP8*	**	n.s.
rEW/ETD-VP4-EW-VP5*-body	**	***
rEW/ETD-VP4-EW-VP5*-foot	**	*

^1.^ Fecal RV shedding curves were compared with either EW or rEW/ETD-VP4 by two-way ANOVA with the Tukey multiple comparison test. ^2.^ Statistical significance is indicated as n.s.: not significant; * *p* < 0.05; ** *p* < 0.01; *** *p* < 0.001; n.a.: not applicable.

## Data Availability

Data related to this paper may be requested from the authors.

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
