# Peer review of "Reverse Genetics of Murine Rotavirus: A Comparative Analysis of the Wild-Type and Cell-Culture-Adapted Murine Rotavirus VP4 in Replication and Virulence in Neonatal Mice"

_viruses, 2024, doi:10.3390/v16050767_

Round 1
Reviewer 1 Report
Comments and Suggestions for Authors
Kawagishi et al used a reverse genetics system to successfully rescue a recombinant RV consisting of all 11 segments of murine origin and constructed a panel of rRVs with the same backbone. The authors then conducted animal studies to examine the differences in replication and virulence between lab adapted strain and non-adapted strain. The manuscript is clear and well structed, with sufficient novelty and significance.
Suggestions:
1. Title: the authors analyzed replication and virulence by animal studies after constructing rRVs. “by reverse genetics” is only a portion of the work. Consider rewrite.
2. The authors used and discussed various strains throughout the manuscript. It would be beneficial to include a table that outlines the differences for the readers' convenience. Is EDIM-EW equivalent to EW? Are there any differences between EW and rD6/2, aside from VP4?
3. It is somewhat surprising that the author managed to rescue all three rRVs, namely rEW/ETD-VP4-EW-VP8*, rEW/ETD-VP4-EW-VP5*-body, and rEW/ETD-VP4-220 EW-VP5*-foot. Have the authors attempted to combine the mutations e.g., rEW/ETD-VP4-220 EW-VP5* to asses a) if they can be rescued and b) the impact on infection and virulence in neonatal mice?
4. Line 190 and 295: consider sequencing the entire plasmids. Given the finicky nature of reverse genetics of RV, a more comprehensive discussion could benefit the scientific community. Do the authors suspect that other factors can contribute to the surprise, such as the passage number of MA104, plasmids prep procedure, etc?
5. Figure. 2. Consider explore methods to conduct statistical analysis. Figure. 2D and E are similar and are more similar to 2A than 2F. More discussion is warranted even without statistical analysis.
6. The material and methods can be improved with additional details. A) Consider adding whole plasmid sequences in addgene and associate sequence names with rRV strains listed in the table suggested above. B) Are Rabbit anti-RV DLP and pig anti-RV TLP commercially available? If so, please provide company and catalog numbers. C) Provide cat# for HRP-conjugated anti-Rabbit IgG polyclonal antibody and ELISA plates.
7. Line 45 Replace Rotateq with RotaTeq.
Author Response
Response: We greatly appreciate the reviewer’s detailed and positive summary of our work. We have now provided additional clarification, as elaborated below.
Comment 1. Title: the authors analyzed replication and virulence by animal studies after constructing rRVs. “by reverse genetics” is only a portion of the work. Consider rewrite.
Response: We thank the reviewer for the suggestion. Although we previously reported the use of reverse genetics for murine-like RV (i.e., rD6/2-2g), this is the first report of the rescue of recombinant RV with all 11 gene segments derived from a bona fide murine RV strain. Therefore, we would like to keep the term “reverse genetics” to highlight the current work. We have changed the title as listed below.
New Title: Reverse genetics of murine rotavirus: Comparative analysis of wild-type and cell culture-adapted murine rotavirus VP4s in replication and virulence in neonatal mice
Comment 2. The authors used and discussed various strains throughout the manuscript. It would be beneficial to include a table that outlines the differences for the readers' convenience. Is EDIM-EW equivalent to EW? Are there any differences between EW and rD6/2, aside from VP4?
Response: We thank the reviewer for making this helpful suggestion. The EDIM-EW RV strain used in the study was the non-tissue-culture-adapted EW strain. Since none of the three VP4 chimeric viruses showed similar fecal RV shedding to the EW strain, our data suggest that there are differences in amino acid sequence in the other 10 gene segments. However, we have not determined the whole sequence of EW strain used in this study. We agreed to include a summary table for the readers’ convenience. We have included an additional table for the gene constellation of the viruses used in this study. Please see line 66.
Table 1. Gene constellation of D6/2 and recombinant viruses.
|
Gene segment (Viral protein) |
D6/2 |
rD6/2-2g |
rEW/ETD-VP4 |
|
Gene segment 1 (VP1) |
EW |
SA11 |
EW |
|
Gene segment 2 (VP2) |
EW |
EW |
EW |
|
Gene segment 3 (VP3) |
EW |
EW |
EW |
|
Gene segment 4 (VP4) |
RRV |
RRV |
ETD |
|
Gene segment 5 (NSP1) |
EW |
EW |
EW |
|
Gene segment 6 (VP6) |
EW |
EW |
EW |
|
Gene segment 7 (NSP3) |
EW |
EW |
EW |
|
Gene segment 8 (NSP2) |
EW |
EW |
EW |
|
Gene segment 9 (VP7) |
EW |
EW |
EW |
|
Gene segment 10 (NSP4) |
EW |
SA11 |
EW |
|
Gene segment 11 (NSP5/6) |
EW |
EW |
EW |
Comment 3. It is somewhat surprising that the author managed to rescue all three rRVs, namely rEW/ETD-VP4-EW-VP8*, rEW/ETD-VP4-EW-VP5*-body, and rEW/ETD-VP4-220 EW-VP5*-foot. Have the authors attempted to combine the mutations e.g., rEW/ETD-VP4-220 EW-VP5* to asses a) if they can be rescued and b) the impact on infection and virulence in neonatal mice?
Response: We thank the reviewer for the suggestion. Despite multiple attempts, we were unable to rescue a full murine rotavirus with all four VP5* mutations.
Comment 4. Line 190 and 295: consider sequencing the entire plasmids. Given the finicky nature of reverse genetics of RV, a more comprehensive discussion could benefit the scientific community. Do the authors suspect that other factors can contribute to the surprise, such as the passage number of MA104, plasmids prep procedure, etc?
Response: We thank the reviewer for the suggestion. Three of the authors (Takahiro Kawagishi, Liliana Sanchez-Tacuba, and Siyuan Ding) prepared the plasmids and attempted to rescue recombinant D6/2 by themselves at different times with different batches of the plasmids but were not successful. However, when we constructed the plasmids again, we could rescue the virus, although we used the same commercially available plasmid prep kit and the same cell lines. Therefore, we suspect that the plasmid clone contributes to the rescue efficiency rather than the condition of MA104 cells and the plasmid prep procedure. The full sequencing of the rescue plasmids may help find the difference between these clones. We have included this point in the discussion.
See lines 303-304: Whole plasmid sequencing of the plasmids would be helpful to examine whether there is any difference between the clones.
Comment 5. Figure. 2. Consider explore methods to conduct statistical analysis. Figure. 2D and E are similar and are more similar to 2A than 2F. More discussion is warranted even without statistical analysis.
Response: We thank the reviewer for the suggestion. Throughout the course of infection, we did not follow the diarrhea incidence of each pup in this experiment, so unfortunately we could not compare the percentage of diarrhea using an area under the curve analysis.
Comment 6. The material and methods can be improved with additional details. A) Consider adding whole plasmid sequences in addgene and associate sequence names with rRV strains listed in the table suggested above. B) Are Rabbit anti-RV DLP and pig anti-RV TLP commercially available? If so, please provide company and catalog numbers. C) Provide cat# for HRP-conjugated anti-Rabbit IgG polyclonal antibody and ELISA plates.
Responses: 6A) We will consider contacting Addgene to deposit the plasmids after the paper is accepted for publication. Addgene will perform full plasmid sequencing as part of their validation protocol.
6B) These antisera were generated in Dr. Harry Greenberg’s lab many years ago and are not commercially available. We have included this information in Materials and Methods.
See lines 174-176: The relative quantity of RV fecal shedding was assessed by sandwich ELISA, as previously described, using guinea pig anti-RV TLP antiserum and rabbit anti-RV DLP antiserum generated in the Greenberg lab.
6C) Thank you for the suggestion. We have included the cat numbers for HRP-conjugated anti-rabbit IgG polyclonal antibody and ELISA plates in the Materials and Methods.
See lines 176-177: ELISA plates (E&K Scientific Products, cat. #EK-25061) were…
Lines 181-182: HRP conjugated anti-rabbit IgG (Sigma-Aldrich, cat. #A0545), and…
Comment 7. Line 45 Replace Rotateq with RotaTeq.
Response: Thank you very much for pointing out our mistake. We have now replaced “Rotateq” with “RotaTeq”.

Reviewer 2 Report
Comments and Suggestions for Authors
Kawagishi et al, present a nicely written manuscript that describes a reverse genetic approach aimed at dissection of the molecular determinants of rotavirus virulence in a mouse model. The work establishes for the first time a recombinant RV with all 11 gene segments of murine origin using a VP4 gene from a cell culture-adapted murine strain. Into this genetic background the authors compare amino acid substitutions derived, separately, from the VP8* and VP5* domains of VP4 from cell culture-adapted and wild-type (virulent) EW strain. Their findings, though modest in quantitative terms, indicate that VP4 chimeric viruses with amino acids changes in VP5* body or foot regions, but not changes in VP8*, slightly increased the amount of RV shedding compared to a virus with ETD-VP4. The study identifies a viable methodology for further investigating amino acid changes that play a role in the molecular mechanism of host range restriction among RVs.
Two minor points;
The authors do not comment on the relative effects of the different VP4 sequences on replication fitness in MA104 cells. Given VP4 plays a major role in adaptation of virus to cell culture did any of the changes in VP8* or VP5* alter RV titers in MA104?
Discussion, page 8 (Line 294)
It suggests that clonal differences might affect rescue efficiency in reverse genetics. It is uncertain whether there is a difference in some other parts of the plasmid, and even if that is the case, whether it affects reverse genetics results.
This seems unnecessarily speculative and somewhat implausible and might be deleted.
Author Response
Response: We greatly appreciate the reviewer’s detailed and positive summary of our work. We have now provided additional clarification, as elaborated below.
Two minor points;
The authors do not comment on the relative effects of the different VP4 sequences on replication fitness in MA104 cells. Given VP4 plays a major role in adaptation of virus to cell culture did any of the changes in VP8* or VP5* alter RV titers in MA104?
Response: We thank the reviewer for this useful suggestion. In this manuscript, we did not compare the growth kinetics of the viruses in vitro and focused on the viral replication in vivo in the mouse intestines. However, it is important to discuss the possibility that the viruses show different growth in MA104 cells. We have now included this point in the discussion.
See lines 339-340: It is possible that rEW/ETD-VP4-EW-VP5*-body and rEW/ETD-VP4-EW-VP5*-foot replicate better than rEW/ETD-VP4 in MA104 cells.
Discussion, page 8 (Line 294)
It suggests that clonal differences might affect rescue efficiency reverse genetics. It is uncertain whether there is a difference in some other parts of the plasmid, and even if that is the case, whether it affects reverse genetics results. This seems unnecessarily speculative and somewhat implausible and might be deleted.
Response: We thank the reviewer for this suggestion, which was raised by reviewer #1. We have now added this point to the Discussion section. Please see our elaborate response to Comment 4 from reviewer #1.
